# Mechanism of BN-Promoting Acicular Ferrite Nucleation to Improve Heat-Affected Zone Toughness of V-N-Ti Microalloyed Offshore Steel

**DOI:** 10.3390/ma15041420

**Published:** 2022-02-15

**Authors:** Zhongran Shi, Tao Pan, Yu Li, Xiaobing Luo, Feng Chai

**Affiliations:** 1Institute of Structural Steels, Central Iron and Steel Research Institute, Beijing 100081, China; luoxiaobing@cisri.com.cn (X.L.); chaifeng@cisri.com.cn (F.C.); 2Yantai CISRI Marine Equipment Materials Applied Technology Corporation Limited, Yantai 265600, China; 3China Iron and Steel Association, Beijing 100711, China; gxpzjsc@163.com

**Keywords:** normalized rolling, microalloyed steel, heat affected zone, acicular ferrite, boron

## Abstract

This study examined the effect of boron (B) on the microstructure and toughness of the simulated coarse-grained heat-affected zone (CGHAZ) of normalized vanadium microalloyed offshore steel by using the welding thermal simulation method under different heat inputs for welding. The results showed that when *t*_8/5_ (the cooling time from 800 to 500 °C) increased from 14 s to 24 s, In the range of *t*_8/5_ of 24–44 s, the impact energy of the CGHAZ rose initially and then remained constant at around 125 J at −40 °C, and dropped to 79 J when *t*_8/5_ increased to 64 s. The particles (Ti,V)(C,N)-BN and BN contributed in the generation of acicular ferrite, which minimized the loss of CGHAZ toughness due to the presence of carbon. Furthermore, the microstructural parameters controlling CGHAZ toughness were the contents of the high misorientation grain boundaries and effective grain size at a tolerance angle of 15° at varied heat inputs.

## 1. Introduction

Marine engineering is a critical component of the oil and gas extraction process, and offshore steel plays a critical role in guaranteeing its safety. Offshore steels must have high strength and toughness, corrosion resistance, fatigue resistance, and be easy to weld due to the long-term operation of marine engineering equipment at low temperatures in the presence of waves, deformation, sea water corrosion, and other complex service variables [1,2,3,4]. With the development of large-scale offshore engineering equipment, the thickness-related specifications of offshore steel are becoming more stringent and the uniformity of its cross-sectional performance is becoming more important. To ensure consistency and stability of performance, normalized rolling high-strength offshore steel with a yield strength of 355 MPa and appropriate toughness at −40 °C is now used for the building of pile structures, such as the jacket platform and tower barrel for offshore wind generation [4,5,6].

Normalized rolling or normalized steel has a higher carbon content/carbon equivalent than steel manufactured using the thermomechanical control process (TMCP), resulting in brittle microstructures such as side-lath ferrite, granular bainite, and Martensite/Austenite island (M-A) and destroying the CGHAZ toughness. Improving the toughness of the heat-affected zone of normalized rolled or normalized offshore steel has become a popular subject of research, and breakthroughs have been made in research on vanadium and nitrogen microalloying-based TMCP steel with a yield strength of 355 MPa and adequate toughness at −20 °C. Jing et al. [7] researched the continuous cooling behavior of CGHAZ in 09MnVTiN steel, and showed that (Ti,V)(C,N) can act as the nucleation core of intragranular ferrite (IGF). Hu et al. [8] found that V-rich caps precipitated from Ti-rich particles and promoted the nucleation of IGF. Zhang et al. [9] studied the microstructure and toughness of the CGHAZ in 0.14C-0.01V-0.0064N-0.17Mo steel, and found that V(C,N), (MnS + VN), and (Ti,V)(C,N) promoted the formation of IGF. Hu [10] investigated the microstructure and toughness of the CGHAZ in 0.12C-0.1V-0.018N steel, where the maximum impact energy was 53 J at −20 °C. Some researchers have shown that B can promote intragranular ferrite nucleation and improve CGHAZ toughness. Shi et al. [11] researched the effect of boron on the intragranular ferrite nucleation mechanism in the CGHAZ of low-carbon V-N-Ti TMCP steel at *t*_8/5_ 180 s, and the results showed that BN formed on the undissolved precipitates to increase the nucleation potency of IGF. Funakoshi found that BN nucleated on the surface of Rare Earth Metals (REM) oxide and promoted IGF nucleation, thus improving CGHAZ toughness, and Ohno et al. [12] found that a “C-poor” region was formed around Fe_23_(B,C)_6_ to increase the transition temperature of ferrite, and promoted the nucleation of GBF.

This study offered data to better understand the effect of B on the microstructures and toughness of CGHAZ in normalized vanadium microalloyed offshore steel with different heat inputs. Precipitation was observed by using the carbon film replica technique and scanning electron microscopy, and the crystallography was analyzed by EBSD. Furthermore, the micromechanisms of fracture were investigated by observing the Charpy impact fracture surface combined with secondary cracks. The relationship among the heat input, microstructural evolution, precipitation, and toughness of the simulated CGHAZ was hence clarified.

## 2. Materials and Methods

### 2.1. Materials

Table 1 shows the chemical composition of the experimental A and B steel, which were melted in a 50 kg vacuum induction furnace. The A steel was utilized to investigate the impact of B on CGHAZ microstructure and toughness, while the B steel served as a contrast steel. The ingots were forged into billets with dimensions of 140 mm × 100 mm × 60 mm (photos of examples can be found in Figure 1a), and then rolled in a two-stage controlled rolling process. The process parameters were as follows: opening rolling at 1200 °C, finishing rolling temperatures of 920 °C and 860 °C, and air cooling after rolling. The thickness of the steel plate was 12 mm, as shown in Figure 1b.

### 2.2. Welding Thermal Simulation Experiment

As illustrated in Figure 1c, the welding thermal simulation sample was cut longitudinally along the rolled steel plate and machined to a size of 10.5 mm × 10.5 mm × 65 mm. The microstructure and toughness of CGHAZ were simulated by the Gleeble 3800 thermal simulator (Dynamic Systems Inc., New York, NY, USA). The Rykalin 3D model was used to simulate the welding thermal cycle of the 80 mm steel plate. The sample was heated at 1250 °C at a rate of 100 °C/s, and was then heated at 50 °C/s to the peak temperature of 1350 °C × 1 s. The cooling times from 800 °C to 500 °C (*t*_8/5_) were 14 s, 24 s, 34 s, 44 s, and 64 s, followed by air cooling to 300 °C; the parameters are shown in Table 2. To observe the evolution of the simulated CGHAZ microstructure during weld cooling, the cooling was interrupted at different temperatures by water quenching to obtain partially transformed microstructures.

### 2.3. Microstructural Characterization

The metallographic sample was cut from the thermocouple of the welding thermal simulation sample, and was etched with a 4 vol.% Nital solution and examined via the Leica MEF4M optical microscope (OM, Wetzlar, Germany). Figure 2a shows how the M-A of simulated CGHAZ were recognized and drawn using Adobe Photoshop, as shown in Figure 2b. Image-Pro Plus was used to measure diameter, maximum diameter, and area percentage of the M-A constituent using SEM micrographs, as shown in Figure 1b. The statistical area had to be at least 81 mm^2^ in size. Samples of electron-backscattered diffraction (EBSD) were electropolished in a 10% solution of perchloric acid and ethyl alcohol. The EBSD maps were scanned by Flamenco software (Oxford Instruments, Oxford, UK) with a step size of 0.3 μm, and were analyzed by HKL-Channel 5 software (Oxford Instruments, Oxford, UK). Tolerance angles of 15° were applied. The samples of replicas of carbon extraction and thin foils were examined in a Tecnai G2 F30 high-resolution transmission electron microscope (TEM, Hillsboro, OR, USA) to observe the precipitates of the simulated CGHAZ. The composition of the precipitation was confirmed by energy-dispersive X-ray spectroscopy (EDX). The impact fracture and secondary cracks of the impact samples were observed by SEM.

### 2.4. CGHAZ Toughness

After welding thermal simulation, the samples were notched at the welding thermocouple and further machined to a standard Charpy V-notch impact sample of size 10 × 10 × 55 mm. The impact tests were conducted at −40 °C on an Instron NI750 NI pendulum impact-testing machine (NCS Ttesting Technology CO., LTD., Beijing, China) using standard Charpy V-notch impact test (GB/T 229-2007).

## 3. Results

### 3.1. Effect of t_8/5_ on Microstructure in Simulated CGHAZ

Figure 3 illustrates the microstructural morphology of the simulated CGHAZ at different values of *t*_8/5_. For A steel, at *t*_8/5_ 14 s, the microstructure consisted of a small amount of grain boundary ferrite (GBF), side-lath ferrite (FSP), Bainite (B), and a small amount of AF, as shown in Figure 3a. At *t*_8/5_ 24 s, the amount of AF, GBF, and FSP increased, whereas the content of B decreased, meanwhile degenerated pearlite (P) formed around GBF, as shown in Figure 3b. On increasing the *t*_8/5_ to 64 s, the number and size of the GBF and AF increased, as shown in Figure 3d–e. For B steel, the microstructure in the simulated CGHAZ consisted of a small amount of GBF, B, and P, as shown in Figure 3f. It can be seen that boron could increase the production of AF, resulting in microstructure differences between the two test steels.

Figure 4 and Table 3 respectively show SEM micrographs and statistical results of M-A content of the simulated CGHAZ in A steel at different *t*_8/5_. At *t*_8/5_ 14 s, the second phase structure was M-A and Fe_3_C, distributing between lath bainite ferrite or needle-like ferrite, as shown in Figure 4a,b. As shown in Figure 4c, when the *t*_8/5_ increased to 34 s, the M-A was distributed between the needle-like ferrite or around GBF boundary while its area fraction decreased from 5.3% to 2.7%. At *t*_8/5_ 44 s, the M-A disappeared, and the second phase structure of CGHAZ was Fe_3_C.

Figure 5 shows image quality maps of grain boundary misorientation at different values of *t*_8/5_; the red lines represent low misorientation boundaries (LMB) of 2°–15°. The black lines stand for high misorientation boundaries (HMB) exceeding 15°. Figure 6 shows the content of the high grain boundaries percent (≥15°) and mean effective size with the tolerance angle 15° as a function of *t*_8/5_. When *t*_8/5_ was 14 s, the HMB were found between blocks of bainitic lath or needle-like ferrite, the content of the HMB was 38%, and LMB were found in the bainitic ferrite. As *t*_8/5_ continued to increase, the content of B decreased while the contents of AF and GBF increased, as shown in Figure 5b–e. HMB increased as well. As *t*_8/5_ reached 64 s, the content of HMB was 53%. As shown in Figure 6a, with the increase of *t*_8/5_, the effective grain size decreased slightly and then increased, as shown in Figure 6b.

Figure 7 shows TEM micrographs of the simulated CGHAZ in A steels at different values of *t*_8/5_. At *t*_8/5_ 14 s, the lath size of bainitic ferrite was about 0.6 μm, as shown in Figure 7a, there were M-A islands distributing between lath bainite ferrite (Figure 7b), and the AF was observed (Figure 7c). As *t*_8/5_ increased to 24 s, AF became coarsened (Figure 7d), the content of M-A islands decreased, and the pearlite structures appeared (Figure 7e). The lath size of bainitic ferrite increased to about 1.5 μm (Figure 7f).

Figure 8 shows the TEM micrographs of precipitates of A steels in the simulated CGHAZ at different values of *t*_8/5_. As *t*_8/5_ 14 s, small and medium (Ti,V)(C,N) particles were observed with the average particle size of 34 nm, as shown in Figure 8a and energy spectrum (A). The number of precipitated particles in the CGHAZ increased with *t*_8/5_, as shown in Figure 8b,c and energy spectrum (B and C).

### 3.2. Effect of Heat Input on CGHAZ Toughness

Figure 9 shows impact energy of the simulated CGHAZ at different values of *t*_8/5_. For A steel, when the *t*_8/5_ increased from 14 s to 24 s, the −40 °C impact energy increased from 125 J to 149 J. With *t*_8/5_ increasing, the impact energy of the CGHAZ remained constant at 125 J. When *t*_8/5_ increased to 64 s, the impact energy of the CGHAZ decreased to 79 J. While, the change of impact energy was complicated for B steel, and the impact toughness of CGHAZ was significantly lower than that of A steel. In a word, it means that the CGHAZ toughness was improved by adding B element.

### 3.3. Nucleation of Intragranular Ferrite in CGHAZ

Figure 10 shows OM micrographs of simulated CGHAZ in A steel at *t*_8/5_ 24 s quenched at different temperatures. The GBF formed at 800 °C, as shown in Figure 10a; the size and number of the GBF increased with decreasing quenching temperature (Figure 10b–h), and intragranular polygonal ferrite formed at 750 °C, as shown in Figure 10c. The phase transformation of the simulated CGHAZ was complete at a quenching temperature of 525 °C while the microstructures were similar to that of the simulated CGHAZ.

Figure 11 and Figure 12 shows the nucleation location of grain boundary ferrite, intragranular polygonal ferrite, and intragranular acicular ferrite around the Martensite (M) by SEM following the welding thermal cycle curve of *t*_8/5_ at 24 s, followed by water quenching from 750 °C to room temperature. As shown in Figure 11, it can be observed that the GBF nucleated with BN particles in the simulated CGHAZ. As shown in Figure 12, the intragranular polygonal ferrite and intragranular acicular ferrite nucleated at the Ti-rich (Ti,V)(C,N)-BN complex particles and BN particles in the simulated CGHAZ specimen. The BN particles and Ti-rich (Ti,V)(C,N)-BN complex particles served as the nucleation position of ferrite.

### 3.4. Relationship between Microstructural Characteristics and Toughness

Figure 13 and Figure 14 illustrate the CGHAZ morphology and macrostructure of A steel after impact testing. The impact fracture was divided into three zones, namely the fibrous zone, the radiation zone, and the shear lip zone. The v-notch zone was stressed by tensile stress during the Charpy impact test; thus, cracks formed following plastic deformation, and then a fibrous zone formed. The cracks propagated quickly in the radial zone until confronting the pressure stress zone, after which the crack propagation was inhibited by the plastic deformation. Thus, the second fibrous zone was produced. The fibrous zone was composed of large and small dimples, as shown in Figure 14a,d,g. The radial zone consisted of small quasi-cleavage facets, many small ductile dimples, and tear ridges, as shown in Figure 14b,e,h. As *t*_8/5_ increased, an increasing trend in cleavage-element size was observed. Within the shear lip zone, the microfractures consisted of small and large dimples that had an elongated shape, as shown in Figure 14c,h,i.

The OM micrographs in Figure 15 illustrate the paths of secondary crack propagation beneath the fracture surface of A steel. According to Figure 15a,b, when FSP and B were encountered, the cleavage crack expanded linearly, but became deflected when AF was encountered. The CGHAZ formed a large number of acicular structures as *t*_8/5_ increased to 24 s, preventing crack propagation as shown in Figure 15c,d. Due to the establishment of the GBF as the path of propagation of the cleavage crack, the toughness of the CGHAZ decreased at *t*_8/5_ 64 s, as shown in Figure 15e. At the same time, the acicular structures can prevent crack propagation, as shown in Figure 15f.

## 4. Discussion

### 4.1. Effect of Boron on Intragranular Ferrite Nucleation in Simulated CGHAZ

Figure 16 shows the precipitation of the experimental steel at different temperatures in equilibrium. The temperature-dependent precipitation of experimental steel is shown in Figure 16a. From (Ti,V)(C,N) to MnS, BN, (V,Ti)(C,N), and AlN, the precipitation temperatures were 1464 °C, 1362 °C, 1267 °C, 991 °C and 696 °C, respectively. The (Ti,V)(C,N) particles were dominated by Ti, as shown in Figure 16b. At the beginning of its precipitation, the particles were dominated by Ti and N. With decreasing temperature, V precipitated with TiN as the core, and the content of Ti in (Ti,V)(C,N) particles decreased as shown in Figure 16b–d. When cooled to the precipitation temperature of (V,Ti)(C,N), the content of V in the (Ti,V)(C,N) particles decreased and that of Ti increased. The main component of the (V,Ti)(C,N) particles was V, as shown in Figure 16c. With decreasing temperature, V and N in the (V,Ti)(C,N) particles decreased slightly, and the content of C increased. For BN precipitation, the elements B and N were present in the range of the precipitation temperature.

During welding heating, when the specimen was heated to a peak temperature of 1350 °C, the (V,Ti)(C,N), BN, and AlN particles completely dissolved, while MnS, (Ti,V)(C,N) dissoluted partly owing to its dissolution temperature being higher than 1350 °C. In the process of welding cooling, when cooling to the precipitation temperature of BN, a part of the precipitated BN attached to the undissolved (Ti,V)(C,N) nucleation precipitation. As the temperature decreased, some BN separated out, which was consistent with the results of observations of the SEM.

In steel with low nitrogen content, B improved the low temperature toughness of the heat-affected zone (HAZ) in the following manner: (1) boron segregation in the grain boundary of austenite prevented ferrite transformation and subsequently reduced ferrite grain size [13,14]. (2) The intragranular ferrite was nucleated at the position of Fe_23_(B,C)_6_ and BN [11,12,15]. (3) On the contrary, the content of solid solution nitrogen in HAZ decreased. However, under the condition of increasing nitrogen, according to the ideal chemical ratio of TiN, Ti would be combined with 29 ppm of nitrogen and boron would be combined with 15 ppm of nitrogen. That is, because nitrogen was sufficient, boron precipitated in the form of BN rather than Fe_23_(B,C)_6_ and Fe_2_B. Oxides, nitrides, sulfides, and complex precipitated particles were some of the recognized precipitated particles that might be used as the core of intragranular ferrite. The nucleation sites of intragranular ferrite were recognized as VN [16], Ti-rich (Ti,V)(C,N) [8,10], TiN-MnS [17], and BN [11,15,17]. Our findings are consistent with those of Funakoshi et al. [18] and Ohno et al. [12].

### 4.2. Influence of Effective Grain Size on Low Temperature Toughness of CGHAZ

The EBSD method can accurately evaluate the effective grain size and reveal the grain-refining mechanism of complex and multiphase microstructures [19]. It is generally accepted that a high misorientation grain boundary (grain orientation angle ≥ 15°) can inhibit/prevent the propagation of cleavage crack while low grain boundaries (2°≤ misorientation grain boundary < 15°) cannot do so [20]. Therefore, the effective grain characteristics have an important effect on the low-temperature toughness of CGHAZ.

The content of the high misorientation grain boundary in the CGHAZ rose from 38% to 49% when *t*_8/5_ was raised from 14 s to 44 s, and its impact energy increased from 125 to 149 J. The low temperature impact energy of the CGHAZ dropped when *t*_8/5_ rose from 44 s to 64 s, even though the content of the high misorientation grain border grew to 53%. This demonstrated that CGHAZ hardness was influenced by various parameters.

Figure 17 depicts the relationship between effective grain size at a tolerance angle of 15° and CGHAZ impact energy. When *t*_8/5_ increased from 14 s to 24 s, the effective grain size of the CGHAZ decreased from 3.9 μm to 3.7 μm, and the toughness increased from 125 J to 149 J, and remained constant at about 125 J from 24 s to 44 s. When *t*_8/5_ increased from 44 s to 64 s, the effective grain size increased, leading to a reduction in CGHAZ toughness. As a result, the microstructural parameters limiting CGHAZ toughness were the content of the high misorientation grain border and effective grain size.

### 4.3. Effect of Effective Grain Size on Critical Cleavage Stress of CGHAZ

The ductile-brittle transition behavior results from the competition between the yield stress and the cleavage fracture stress, and the ductile-brittle transition temperature is the equilibrium dot between cleavage fracture stress and yield stress [21]. Normally, the fracture stress is independent of temperature, while the yield strength increases rapidly with decreasing temperature. The fracture surfaces will reveal dimples when the fracture stress is higher than the yield strength. When the fracture stress is lower than the yield strength, the fracture surfaces will reveal cleavage. In general, the cleavage fracture stress can be expressed by the following formula [22,23]:(1)σcf=4Eγπ1−ν2C012
where E is Young’s modulus, γ is plastic deformation energy, ν is Poisson’s ratio, and *c*_0_ is the effective grain size.

The *σ*_cf_ is temperature independent, and is determined by such microstructural factors as the grain size of ferrite, effective grain size with an orientation difference of 15°, or the distribution of carbides. In the research work of this paper, when *t*_8/5_ increased from 14 s to 24 s, the *c*_0_ decreased from 3.9 μm to 3.7 μm, leading to *σ*_cf_ increasing, leading to a reduction in CGHAZ toughness. When *t*_8/5_ increased from 34 s to 64 s, the effective grain size increased (*σ*_cf_ decreased), leading to a reduction in CGHAZ toughness.

## 5. Conclusions

The precipitation characteristics, microstructural evolution, effective grain size, and CGHAZ toughness of the simulated CGHAZ in normalized V-N-Ti microalloyed offshore steel subjected to different values of *t*_8/5_ were investigated here. The major conclusions can be summarized as follows:(1)The impact energy of the CGHAZ at −40 °C grew initially and then remained constant at about 125 J in the range of *t*_8/5_ of 24 s–44 s, then reduced to 79 J when *t*_8/5_ was raised to 64 s.(2)The microstructural parameters controlling CGHAZ toughness were effective grain at various welding heat inputs.(3)The production of acicular ferrite was facilitated by the (Ti,V)(C,N)-BN and BN particles, which minimized the degrading effect of carbon on CGHAZ toughness.

## Figures and Tables

**Figure 1 materials-15-01420-f001:**
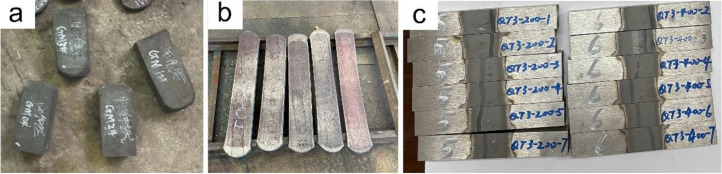
Specimen photographs (**a**) billets; (**b**) experimental plates; (**c**) welding thermal simulation specimens.

**Figure 2 materials-15-01420-f002:**
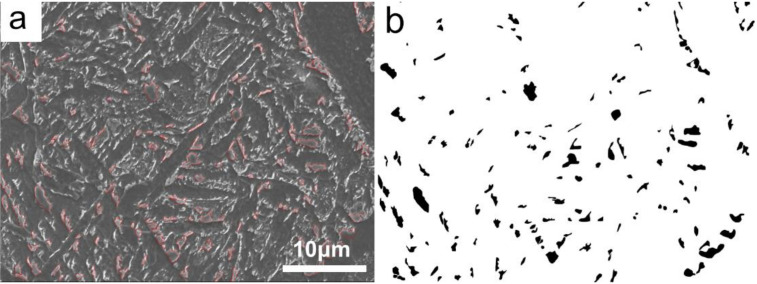
The statistical method of M-A in simulated CGHAZ. (**a**) the morphology of M-A island; (**b**) M-A were recognized and drawn using Adobe Photoshop.

**Figure 3 materials-15-01420-f003:**
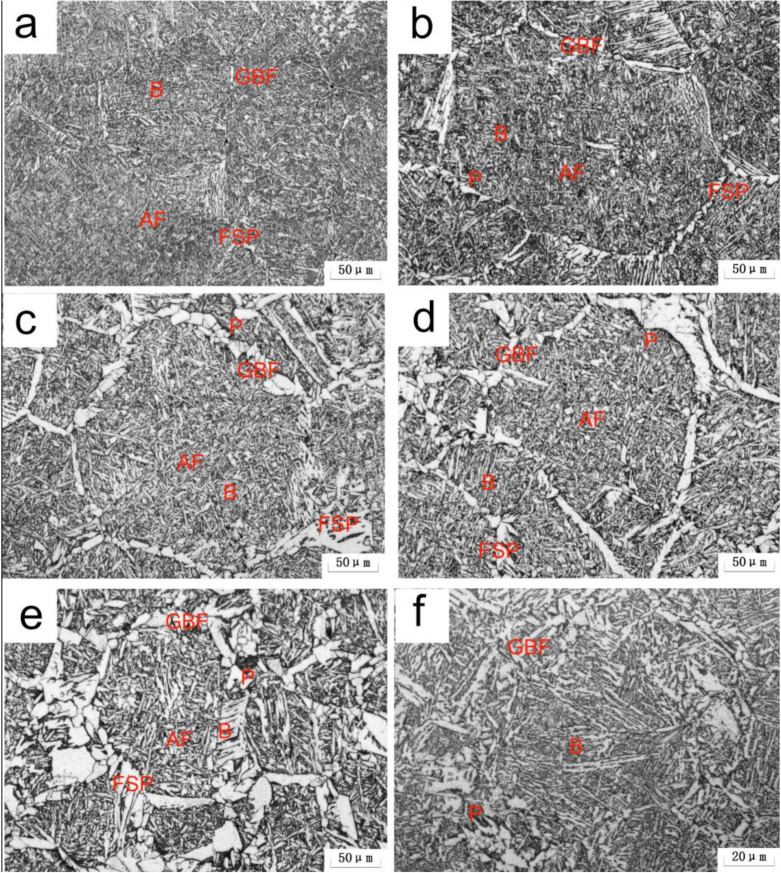
OM micrographs of simulated CGHAZ at various *t*_8/5_ (**a**–**e**) A steel, (**f**) B steel; (**a**) 14 s, (**b**,**f**) 24 s, (**c**) 34 s, (**d**) 44 s, (**e**) 64 s.

**Figure 4 materials-15-01420-f004:**
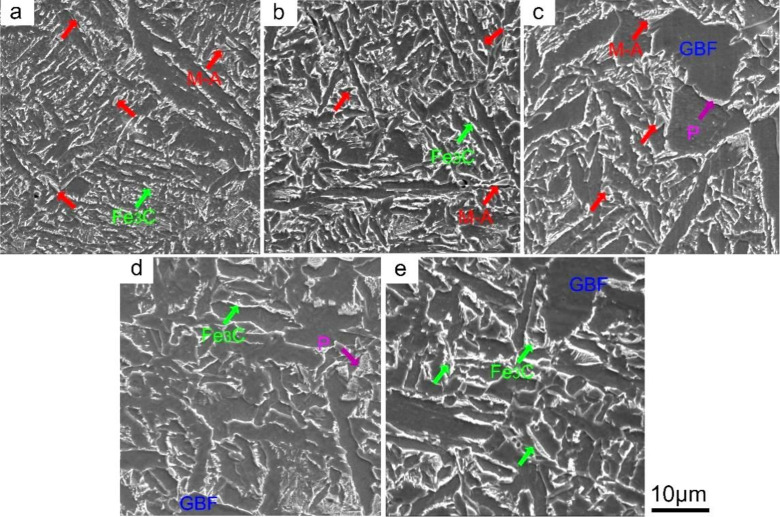
SEM micrographs of simulated CGHAZ in A steel at various *t*_8/5_. (**a**) 14 s, (**b**) 24 s, (**c**) 34 s, (**d**) 44 s, (**e**) 64 s.

**Figure 5 materials-15-01420-f005:**
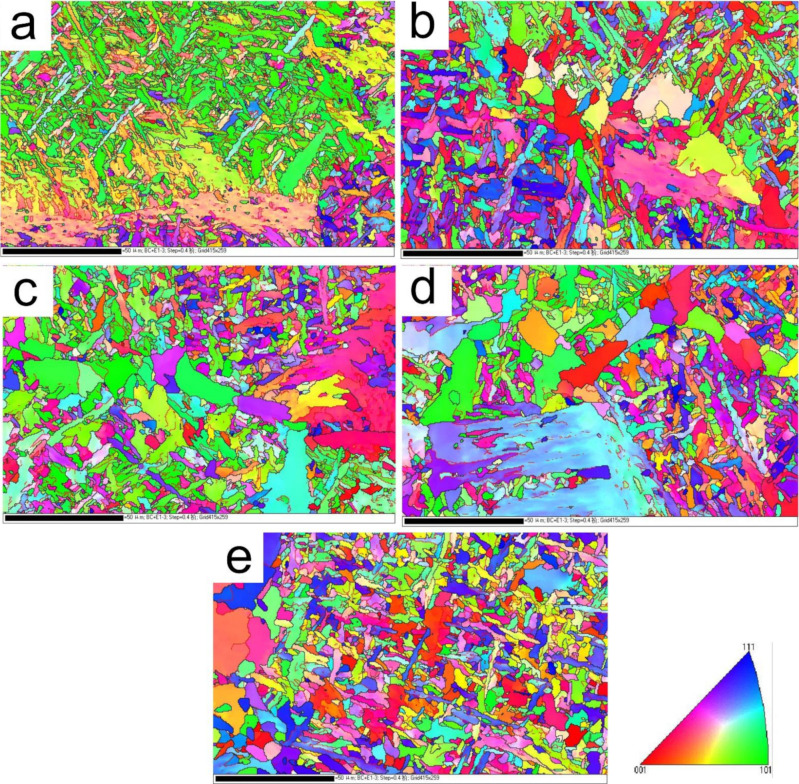
The EBSD orientation images with grain boundary misorientation distribution of A steel at different *t*_8/5_ (**a**) 14 s, (**b**) 24 s, (**c**) 34 s, (**d**) 44 s, (**e**) 64 s.

**Figure 6 materials-15-01420-f006:**
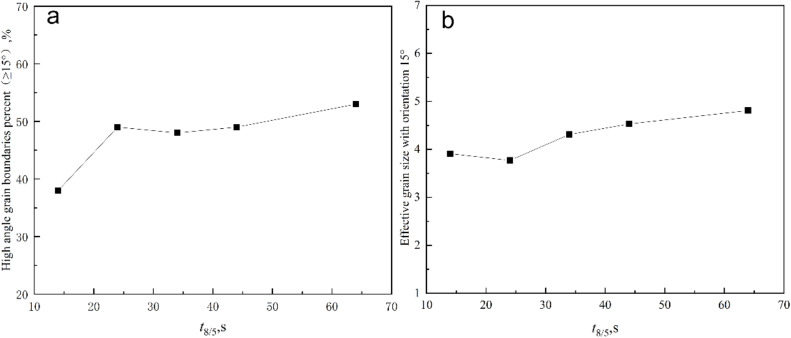
The content of high grain boundaries (≥15°) as a function of *t*_8/5_ (**a**); EBSD effective size with the tolerance angle 15° as a function of *t*_8/5_ (**b**).

**Figure 7 materials-15-01420-f007:**
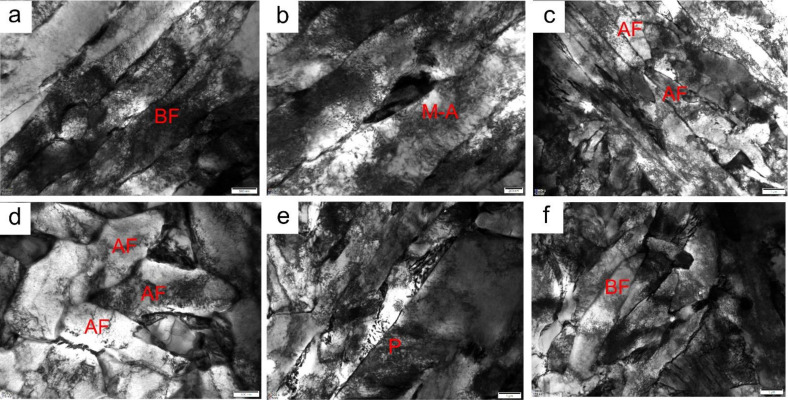
TEM micrographs of simulated CGHAZ in A steel (**a**–**c**) *t*_8/5_ 14 s; (**d**–**f**) *t*_8/5_ 24 s.

**Figure 8 materials-15-01420-f008:**
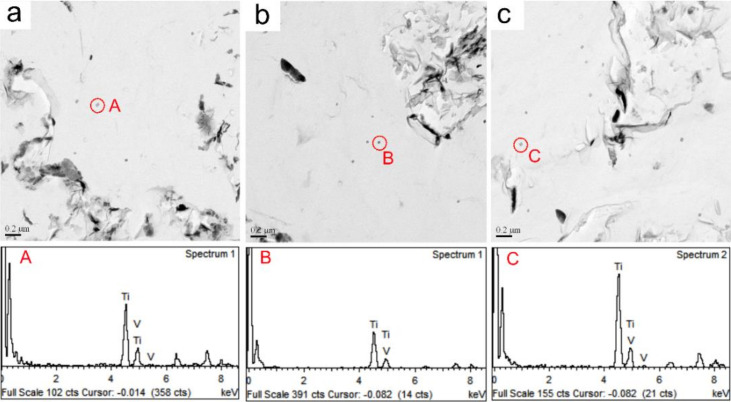
TEM carbon replica analysis micrographs of particles in simulated. CGHAZ of A steel (**a**) *t*_8/5_ 14 s, (**b**) *t*_8/5_ 34 s, (**c**) *t*_8/5_ 64 s. (A–C) the energy spectrum of (Ti,V)(C,N) particles.

**Figure 9 materials-15-01420-f009:**
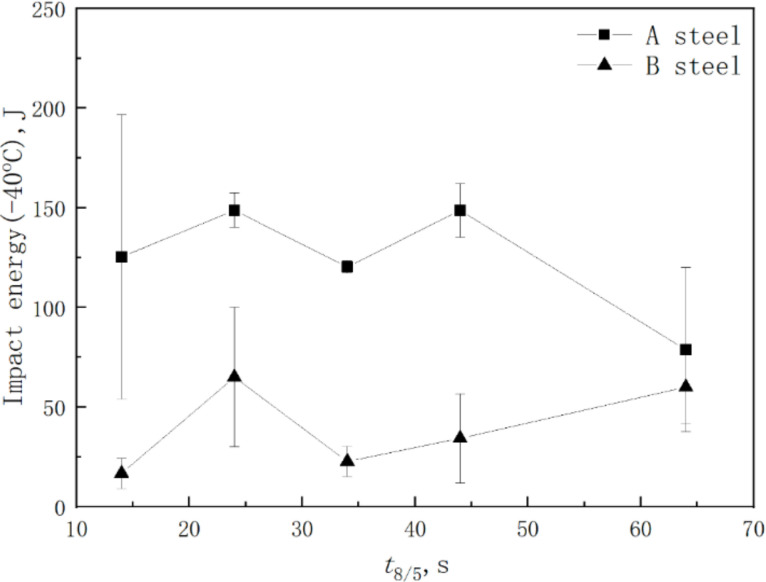
The impact energy of simulated CGHAZ at −40 °C as a function of different *t*_8/5_.

**Figure 10 materials-15-01420-f010:**
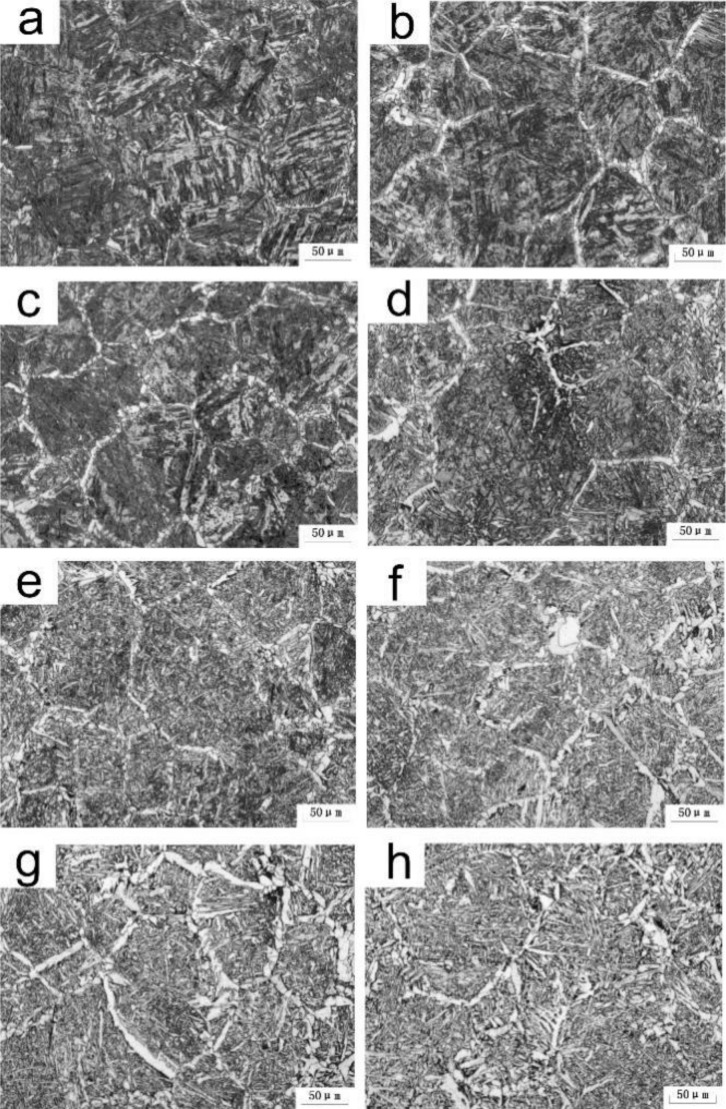
OM micrographs in simulated CGHAZ of A steel at *t*_8/5_ 24 s quenched at the different temperatures of (**a**) 800 °C, (**b**) 775 °C, (**c**) 750 °C, (**d**) 725 °C, (**e**)700 °C, (**f**) 600 °C, (**g**) 550 °C, (**h**) 500 °C.

**Figure 11 materials-15-01420-f011:**
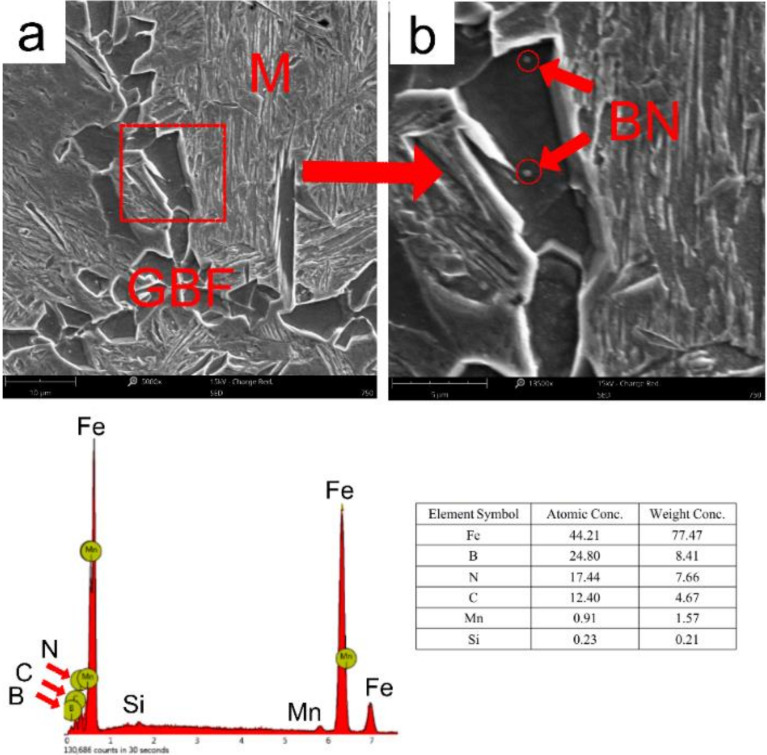
SEM micrographs of GBF nucleated at BN particles in simulated CGHAZ specimen of A steel following the welding thermal cycle curve of *t*_8/5_ 24 s, and then water quenching at 750 °C to room temperature. (**a**) shows the morphology with low multiples; (**b**) is partial enlargement of (**a**).

**Figure 12 materials-15-01420-f012:**
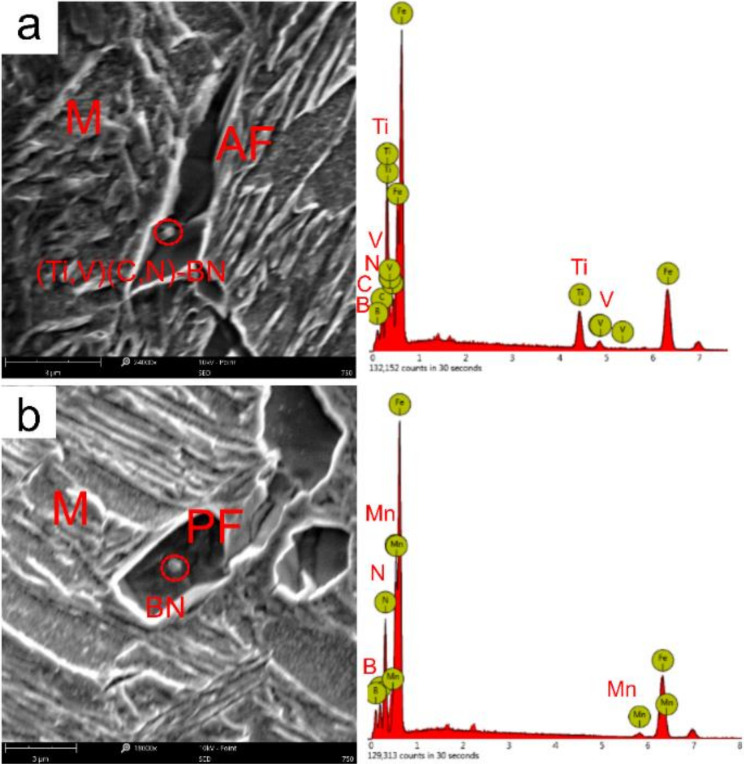
SEM micrographs of intragranular polygonal ferrite and intragranular acicular ferrite nucleated at Ti-rich (Ti,V)(C,N)-BN complex particles (**a**) and BN particles (**b**) in simulated CGHAZ specimen following the welding thermal cycle curve of *t*_8/5_ 24 s, and then water quenching at 750 °C to room temperature.

**Figure 13 materials-15-01420-f013:**
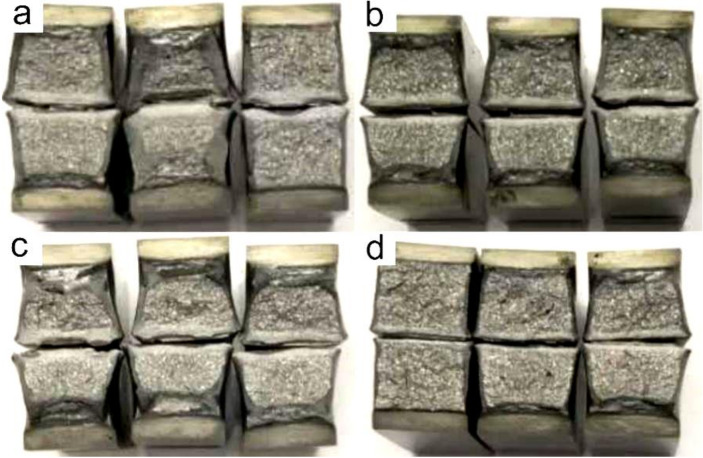
The macrostructure of A steel in simulated CGHAZ after impact testing. (**a**) *t*_8/5_ 14 s; (**b**) *t*_8/5_ 34 s; (**c**) *t*_8/5_ 44 s; (**d**) *t*_8/5_ 64 s.

**Figure 14 materials-15-01420-f014:**
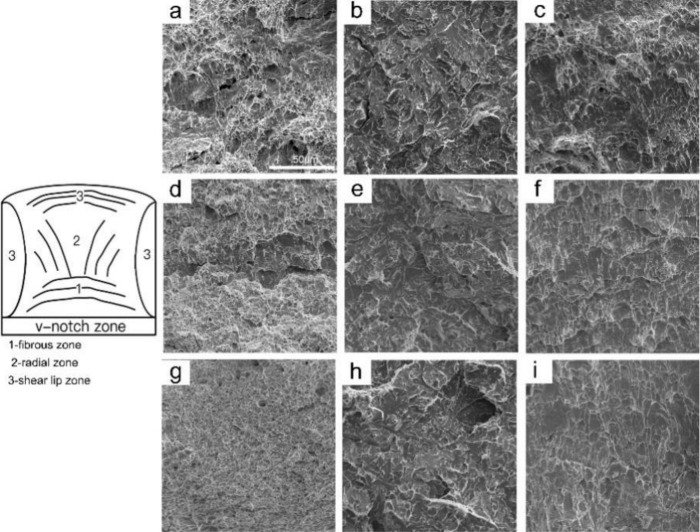
SEM morphology observation of A steel in the simulated CGHAZ impact test specimen at different *t*_8/5_ (**a**–**c**) *t*_8/5_ 14 s; (**d**–**f**) *t*_8/5_ 34 s; (**g**–**i**) *t*_8/5_ 64 s; (**a**,**d**,**g**) fiber zone; (**d**,**e**,**h**) radial zone; (**c**,**f**,**i**) shear lips zone.

**Figure 15 materials-15-01420-f015:**
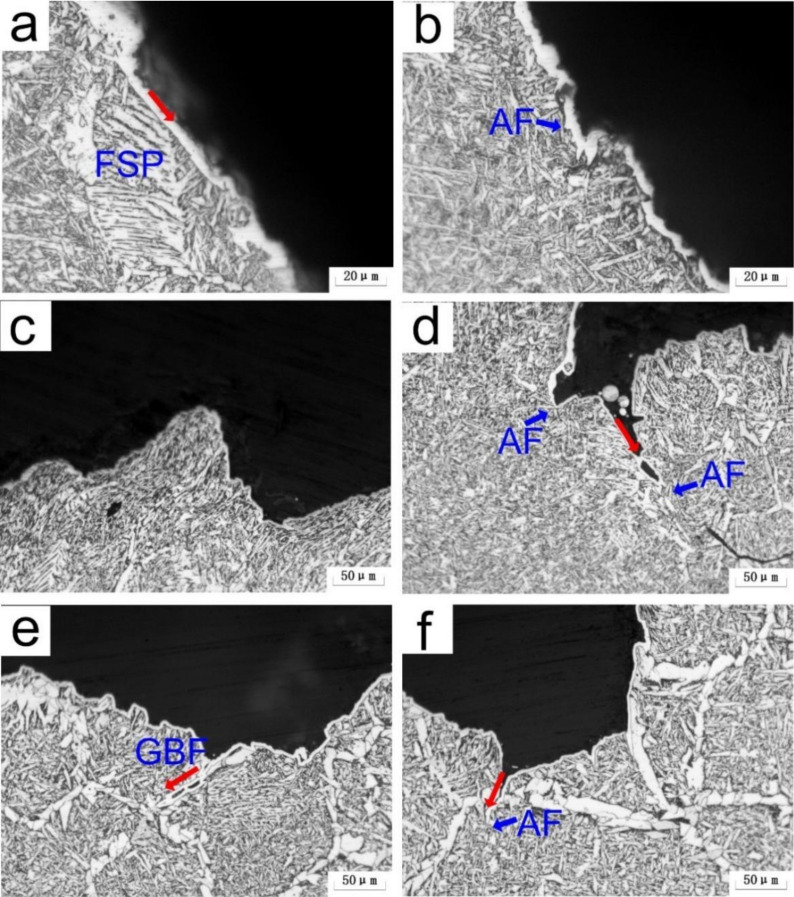
OM micrographs showing secondary cracks propagation paths underneath the fracture surface of specimens radical zone in A steel (**a**,**b**) *t*_8/5_ 14 s; (**c**,**d**) *t*_8/5_ 24 s; (**e**,**f**) *t*_8/5_ 64 s.

**Figure 16 materials-15-01420-f016:**
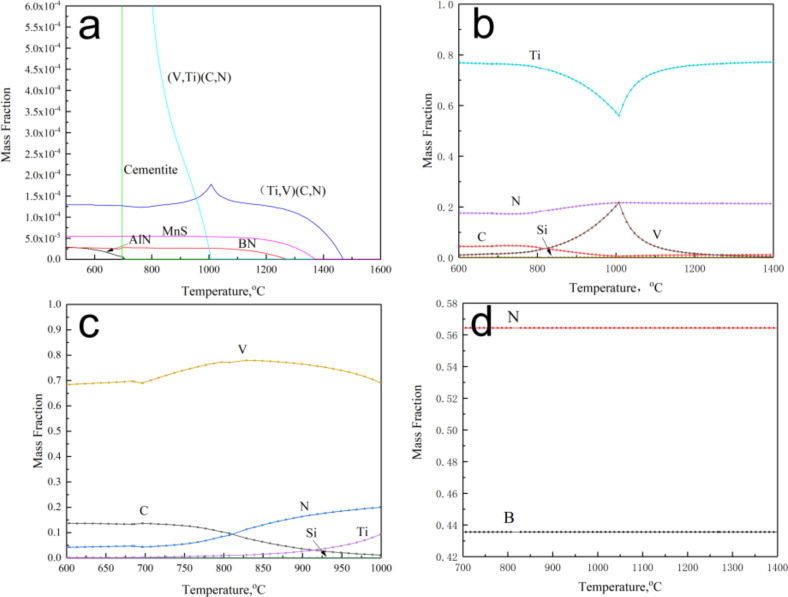
Particles precipitated in equilibrium state of A steel. (**a**) the temperature-dependent precipitation of experimental steel; (**b**) the composition of (Ti,V)(C,N) particles changes with temperature; (**c**) the composition of (V,Ti)(C,N) particles changes with temperature; (**d**) the composition of BN particles changes with temperature.

**Figure 17 materials-15-01420-f017:**
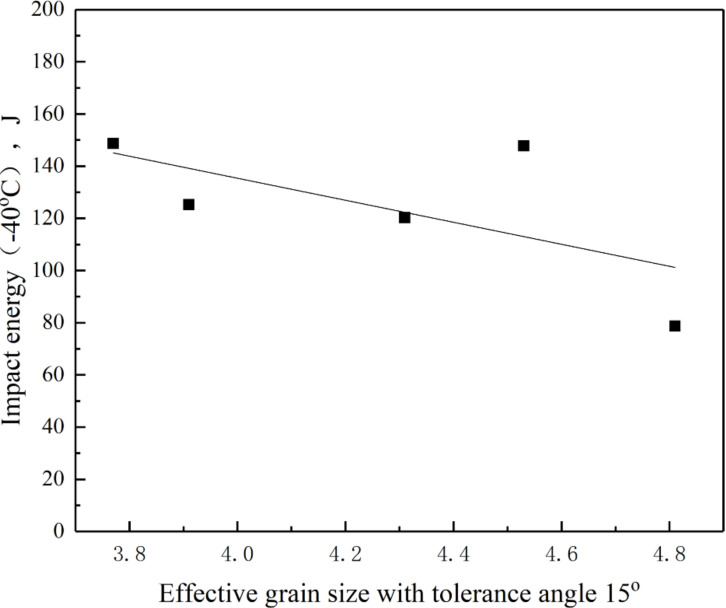
Impact energy in simulated CGHAZ of A steel varied with effective grain size at a tolerance angle of 15°.

**Table 1 materials-15-01420-t001:** Chemical composition of the experimental steels (wt.%).

Steel	C	Si	Mn	S	P	V	Ti	B	N
A	0.14–0.17	0.20	1.50	0.002	0.005	0.04–0.10	0.005–0.015	5–30 ppm	50–100 ppm
B	0.14–0.17	0.20	1.50	0.002	0.005	0.04–0.10	0.005–0.015	/	50–100 ppm

**Table 2 materials-15-01420-t002:** Parameters for welding thermal simulation.

Parameters	Values
Simulated plate thickness, mm	80
Density, g/cm^3^	7.8
Specific heat, J/g·°C	0.7
Thermal conductivity, J/cm·s·°C	0.5
Heating rate, °C/s	100 °C
Peak temperature, °C	1350
Holding time, s	1
*t*_8/5_, s	14, 24, 34, 44, 64
Post-heat temperature, °C	25

**Table 3 materials-15-01420-t003:** The statistical results of M-A content in CGHAZ of A steel.

*t* _8/5_	Average Diameter/μm	Maximum Diameter/μm	Area Fraction/%
14 s	0.7	1	5.3
24 s	1.1	1.4	3.1
34 s	1	1.3	2.7
44 s	0	0	0
64 s	0	0	0

## Data Availability

The data presented in this study are available upon request from the corresponding author.

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
