# Peer review of "Mechanism of BN-Promoting Acicular Ferrite Nucleation to Improve Heat-Affected Zone Toughness of V-N-Ti Microalloyed Offshore Steel"

_materials, 2022, doi:10.3390/ma15041420_

Round 1

Reviewer 1 Report

Investigation of steels of offshore gas pipelines, and especially their welds, is an important scientific and practical task.

  1. I invite the authors to expand the introduction and description of the problems of offshore welded gas pipelines with reference to the article by Prof. Poberezhnyi: https://www.sciencedirect.com/science/article/abs/pii/S1644966516300048
  2. The article does not describe the cleavage fracture stress test method used by the authors. I know of several methods, for example, to obtain guaranteed brittle fracture, some authors use nitrogen, see, for example, the article by Prof. Yasniy about toughness and failure (in sciencedirect.com). And how the authors of this article got this parameter, it is necessary to clarify.
  3. In fig. 8 shows TEM carbon replica analysis micrographs of particles, but chemical analysis of local zones cannot be seen. These images are very small. I propose to show them separately.
  4. In fig. 9 shows the impact energy of simulated CGHAZ at -40oC, but I recommend:

4.1. Describe the designation "black squares" and "black triangles" in the title of the picture; without this description, the meaning of the picture is incomprehensible.

4.2. Typically, data for impact strength at -40 C is shown in conjunction with data for a temperature of 20 C. Such a comparison allows you to determine the embrittlement.

  1. In fig. 11, no chemical elements are visible, but a Chinese text is visible in the table. It needs to be replaced with English.
  2. in fig. 12. chemical elements are not visible, the font is very small.
  3. In section 3.4 the authors write that they are investigating toughness, but in fact it is impact toughness. It is necessary to clarify this term.

Scale marks in fig. 13 are very small. The magnification cannot be seen. Analysis of Fig. 13 is very short, I suggest you read the article "Impact toughness of 12Cr1MoV steel" (in sciencedirect.com). It will help the authors analyze micromechanisms of fracture deeper.

Reviewer 2 Report

The manuscript materials-1518507 Mechanism of BN promoting acicular ferrite nucleation to improve heat affected zone toughness of high carbon steel presents the investigation on microstructural evolution, strength properties and fracture of the simulated heat-affected zone in normalized V-N-Ti-B micro-alloyed offshore steel subjected to different cooling conditions. Such steels and their welding processing are very important for many applications in the offshore sector in which high toughness and cracking resistance has to be supported. The Authors present the most essential microstructure properties of micro-alloyed steels and describe the type of precipitates in low-carbon micro-alloyed steels and their influence on the toughness properties. However, the main result for which the article stands for is a simulation of the coarse-grained heat-affected zone (CGHAZ) after welding and cooling at different variants. The results showed that when t8/5 (the cooling time from 800 to 500 oC) increased from 14 s to 24 s, the impact energy of the CGHAZ at -40 oC became about 149 J and decreased to 79 J when t8/5 increased to 64 s. The influence of alloying elements on the CGHAZ formation is also presented. It was stated that the (Ti, V)(C, N)-BN and BN compounds promoted the formation of acicular ferrite, which reduced the deteriorating effect of carbon on CGHAZ toughness. Authors based on the LM, SEM, SEM/EDS/EBSD, TEM and Charpy V-notch impact test observed the microstructure, grain boundaries misorientation and estimated impact strength. The content of the high misorientation grain boundary and effective grain size at a tolerance angle of 15o at different heat areas were the microstructural factors controlling CGHAZ toughness and cleavage fracture stress. The fracture analysis with SEM was also realized. As the mentioned manuscript is important for offshore industry steels microstructural tailoring procedures and the welding process of such materials. However, to attract the Readers' attention in such a form, the Authors should improve the presentation of the results. Also, the poor writing of the manuscript sometimes makes the logical progression of the paper hard to follow. As I read the manuscript, I found some mandatory errors that the authors should correct before the publication. I recommend performing a major revision of the manuscript. Some detailed suggestions are listed in the following points:

  1. I propose to change the title of the article, Is that really a high carbon steel ?, I think it better to say …to improve heat affected zone toughness of V-N-Ti-B micro-alloyed offshore steel.
  2. Keywords should be changed to the words that more precisely relates to the issues discussed in the article i.e. Normalized rolling, micro-alloyed steel, heat affected zone etc.
  3. The Introduction section of the article should be It is only a presentation of the influence of alloying elements in offshore steel. It would be also good to present some applications of such steels and show some of their main forming methods (normalized rolling), strength properties, TTT curve, heat treatment and welding processes parameters. It would be also worth better highlighting the need for such experiments. Try to answer the question Why do you investigate heat affected zone? Why is it important?
  4. Authors should more clearly present the CGHAZ simulation methodology and write some more information on realized welding and post heat treatment after the joining process welding. How treatment would it affect the structure, grain size, mechanical properties of the steels? It would be good to present the simulation parameters scheduled in the table.
  5. It would be also good to show the samples to present the macrostructure of the fracture after the impact test (i.e. supplement Figure 13)
  6. There are some abbreviations that are not clearly explained in the article, when are they used for the first time in the text (M-A, REM, GBF, AF), please correct them.
  7. Figures and figures captions. The pictures are poor quality and small size (Figure 1, 3, 4, 7, 8, 11, 12, 15, 16 EDS chemical analysis,  the scale bars are not clearly visible, please correct figures and some captions for better clarity
  1. The table in Figure 11 should be in English
  2. Sections and subsections should be spaced
  3. Some sentences or words are unclear, please re-edit the fragments and sentences.
  4. Please add space between the units and write the units correctly, (corrections needed in the whole article)
  5. Please start sections with the text, not figures (section 3.3., 4.1., )
  6. Please write references correctly i.e. Line 273 should be [11,12,15] as in Line 281 (please check and correct the whole article)
  7. Dimensions: 140×100×60 mm3, should be 140 x 100 x 60 mm (please check the whole article)
  8. The conclusions in the article are too general. I suggest re-editing the conclusions and extending the description of the applications for the future here.

There are no more comments that I felt to comment on. In general, the article requires comprehensive editorial refinement and proofreading at many points. English is unclear and hard to read in some fragments. Some sentences are unclear please re-edit the fragments and sentences for better clarity. The issues presented in the article are suitable for publication in the Journal Materials, however, to increase their scientific value article requires the addition of more images and should be again analyzed by the Authors. I also want to point out that the Authors should format the manuscript and according to the journal's and the MDPI publishing house guidelines.
I recommend the paper for publication in Journal Materials after major revisions

Reviewer 3 Report

This paper evaluates the toughness of the simulated coarse-grained heat affected zone by the welding under different initial conditions on heating and cooling time. The Charpy impact tests are conducted to understand the relationships between the cooling time and impact energy.

Totally, it seems that the procedures and discussions are not clear. The more detail explanations are required.

Some questions and comments are listed below.

  1. The photographs of specimens and experimental procedures should be shown. I could not understand the dimension of the specimen is following any standards or not.
  2. The specimen deformed and fractured in a non-uniform manner. Related to the observation of micrographs, the phenomena will be different if the position of the observations is going to be slightly different. I could not understand which parts of the specimens the authors observed. Especially, the positions or area concerning with CGHAZ and secondary crack shown in Figures 13 and 14 are the same. I cannot understand their correspondence. The authors can prove all the specimens made from the ingot with homogenized microstructures in CGHAZ or not.
  3. The equation on page 13 is not clear. This means the authors think the impact energy shown in Figure 16 and the cleavage fracture stress are irrelevant. 14Jm^-2 of \gamma is referred from the references. However, this should be similar value to the impact energy. The impact energy shows a decrease constantly. The cleavage stress over 1000MPa is so huge compared with impact energy. I think the evaluation method of fracture and discussions in this paper are inappropriate.

For the minor comments, the followings are available.

  1. Some characters in the sentence are covered by the Table 2.
  2. In Table on Figure 11, Chinese characters are included.
  3. Totally, figures are small and unclear. Especially, it is impossible to look at the figures on the micrograph such as Figure 8. As well as the diagrams on the micrographs, the spot encompassed by red circles are so unclear. Figures 4, 5, 8, 11, 12, 15, 16 and 17 are so small. Please care about the legends.
  4. The equation on the page 13 has no numbers. Even though there is only one equation, the numbering is necessary.

Round 2

Reviewer 1 Report

Accept.

Author Response

Thank you for your valuable comments on this article.

Reviewer 2 Report

Manuscript materials-1518507 Mechanism of BN promoting acicular ferrite nucleation to improve heat affected zone toughness of high carbon steel; explores presents the investigation on microstructural evolution, strength properties and fracture of the simulated heat-affected zone in normalized V-N-Ti-B micro-alloyed offshore steel subjected to different cooling conditions. The authors of the article made corrections as suggested by the reviewer and more clearly presented the topic, which is appreciated by the reviewer. Generally, the subject of the manuscript is important for many applications in the offshore sector and joining the micro-alloyed steel elements where high toughness and cracking resistance has to be supported. Such research has a significant scientific value. In general, the article was upgraded and corrected by the Authors. The scientific goal is now more covered by the obtained results. Nevertheless, I have one comment which still needs to be improved. I still propose performing minor editorial revision of the manuscript at some points (Figure captions, Eliminate Beginning of the sections with figures, Some sentences or wording are unclear and should be corrected) and accepting the paper for publication in Journal Materials.

Author Response

I have carefully revised the article and marked the change by yellow.

Reviewer 3 Report

Thank you very much for the authors' great efforts.

I understood them.

I recommend this version for a publication to Materials.

Author Response

(The authors gave the same response as above.)
